

# Cystatin C: immunoregulation role in macrophages infected with *Porphyromonas gingivalis*

Blanca Esther Blancas-Luciano[1,2], Ingeborg Becker-Fauser[3], Jaime Zamora-Chimal[3], Luis Jiménez-García[4], Reyna Lara-Martínez[4], Armando Pérez-Torres[5], Margarita González del Pliego[6], Elsa Liliana Aguirre-Benítez[6] and Ana María Fernández-Presas[2,7]

[1] Posgrado en Ciencias Biológicas, Unidad de Posgrado, Circuito de Posgrados, Ciudad Universitaria, Universidad Nacional Autónoma de México, Ciudad de México, Mexico
[2] Departamento de Microbiología y Parasitología, Facultad de Medicina, Ciudad Universitaria, Universidad Nacional Autónoma de México, Ciudad de México, Mexico
[3] Unidad de Investigación en Medicina Experimental, Hospital General de México, Universidad Nacional Autónoma de México, Mexico City, Mexico
[4] Departamento de Biología Celular. Facultad de Ciencias, Ciudad Universitaria, Universidad Nacional Autónoma de México, Ciudad de México, México
[5] Departamento de Biología Celular y Tisular, Facultad de Medicina, Universidad Nacional Autónoma de México, Mexico City, Mexico
[6] Departamento de Embriología, Facultad de Medicina, Universidad Nacional Autónoma de México, Ciudad de México, Mexico
[7] Centro de Investigación en Ciencias de la Salud, Huixquilucan, Universidad Anáhuac, Estado de México, México

Corresponding author
Ana María Fernández-Presas, presas@unam.mx

## ABSTRACT

**Background**. Periodontitis is a chronic infectious disease, characterized by an exacerbated inflammatory response and a progressive loss of the supporting tissues of the teeth. *Porphyromonas gingivalis* is a key etiologic agent in periodontitis. Cystatin C is an antimicrobial salivary peptide that inhibits the growth of *P. gingivalis*. This study aimed to evaluate the antimicrobial activity of this peptide and its effect on cytokine production, nitric oxide (NO) release, reactive oxygen species (ROS) production, and programmed cell death in human macrophages infected with *P. gingivalis*.

**Methods**. Monocyte-derived macrophages generated from peripheral blood were infected with *P. gingivalis* (MOI 1:10) and stimulated with cystatin C (2.75 µg/ml) for 24 h. The intracellular localization of *P. gingivalis* and cystatin C was determined by immunofluorescence and transmission electron microscopy (TEM). The intracellular antimicrobial activity of cystatin C in macrophages was assessed by counting Colony Forming Units (CFU). ELISA assay was performed to assess inflammatory (TNFα, IL-1β) and anti-inflammatory (IL-10) cytokines. The production of nitrites and ROS was analyzed by Griess reaction and incubation with 2′,7′-dichlorodihydrofluorescein diacetate (H$_2$DCFDA), respectively. Programmed cell death was assessed with the TUNEL assay, Annexin-V, and caspase activity was also determined.

**Results**. Our results showed that cystatin C inhibits the extracellular growth of *P. gingivalis*. In addition, this peptide is internalized in the infected macrophage, decreases the intracellular bacterial load, and reduces the production of inflammatory cytokines and NO. Interestingly, peptide treatment increased ROS production and substantially decreased bacterial-induced macrophage apoptosis.

**Conclusions**. Cystatin C has antimicrobial and immuno-regulatory activity in macrophages infected with *P. gingivalis*. These findings highlight the importance of understanding the properties of cystatin C for its possible therapeutic use against oral infections such as periodontitis.

## INTRODUCTION

Periodontitis is a chronic inflammatory disease that progressively affects the integrity of tooth-supporting tissues (*Hajishengallis & Chavakis, 2021*). Although periodontitis has a multifactorial etiological factor, including local, systemic, and genetic factors, *Porphyromonas gingivalis* has long been described as the main etiological agent involved in the pathogenesis of this disease, in 75.8% of patients with periodontitis (*Rafiei et al., 2017*). Furthermore, it has been reported that *P. gingivalis* induces experimental periodontitis in the murine model (*Hariyani et al., 2021*).

 *P. gingivalis* is a gram-negative rod-shaped, anaerobic, strictly facultative, and asaccharolytic bacterium (*Smalley & Olczak, 2017*). During the subgingival sulcus infection, *P.gingivalis* expresses various virulence factors such as cysteine proteases, known as gingipains, hemagglutinins, lipopolysaccharide (LPS), nucleoside-diphosphate-kinase (NDK), and fimbriae (*Mendez et al., 2019*; *Jia et al., 2019*; *Atanasova et al., 2016*), which contribute to host-invasion, intracellular persistence, induction of the inflammatory response, and apoptosis of the host cells (*Stathopoulou et al., 2009*). The clinicopathological consequence is inflicting irreversible damage to the periodontal connective tissues (*Sun et al., 2021*; *Lam et al., 2016*). Histopathology of gingiva samples and analysis of gingival crevicular fluid from patients with chronic gingivitis suggest that macrophages are pivotal in the development of a long-lasting inflammatory microenvironment (*Fageeh, Fageeh & Patil, 2021*).

 The inflammatory infiltrate in patients with periodontitis includes 5 to 30% of macrophage populations (*Papadopoulos et al., 2017*). These cells play different roles during the different stages of the disease. The chronic phase is associated with the presence of M1 and M2 phenotypes in the inflammatory infiltrate, being M1 macrophages the most abundant cells (*Blancas-Luciano et al., 2023*; *Yang et al., 2018*). Circulating blood monocytes reach the inflammatory infiltrate and differentiate into M1 macrophages, induced by cytokines such as IFN-γ and TNF-α, or by direct stimulation of TLRs, exposed to pathogen-associated molecular patterns (PAMPs) (*Fageeh, Fageeh & Patil, 2021*). LPS from *P. gingivalis* promotes M1 macrophage differentiation (*Liu et al., 2020*; *Holden et al., 2014*), inducing the production of inflammatory mediators such as IL-1β, IL-6, IL-8, TNF-α, and the inducible nitric oxide synthase (iNOS) (*Parisi et al., 2018*). These mediators amplify the inflammatory response by increasing local blood flow, favoring leukocyte recruitment, osteoclast activation, and subsequent bone loss (*Pignatelli et al., 2020*; *Gurav, 2014*).

*P. gingivalis* has evolved abilities to evade or subvert macrophages immune response. This bacterium promotes the polarization of macrophages and inflammasome activation in different microenvironments (*Lin et al., 2022*). Additionally, *P. gingivalis* uses macrophages as host cells, which allows it to survive intracellularly. Bacterial virulence factors, such as fimbriae, contribute to this phenomenon. Bacterial fimbriae are recognized by TLR2, which induces the activation of a Myd88-independent pathway, into which the Mal/TIRAP/adapter protein is recruited. This pathway prevents phagosome-lysosome fusion in macrophages, promoting the survival of internalized bacteria (*Makkawi et al., 2017*). Subsequently, macrophages and lymphocytes produce matrix metalloproteinases (MMPs), such as MMP-1, MMP-13, MMP-8, and MMP-9, and inflammatory factors, such as IL 1-β and TNF-α (*Parkhill et al., 2000*). All these molecules, and the increase in oxidative stress due to the production of ROS, and the others mentioned above, have a critical role in periodontal tissue destruction (*Song, Jeon & Kim, 2021*; *Cheng et al., 2020*).

Furthermore, diverse cytokines produced by macrophages promote the development of periodontitis. Inflammasome activation induces the maturation of two vital pro-inflammatory cytokines, IL-1α and IL-18 (*Champaiboon et al., 2014*). The upregulation of the inflammasome components caspase-1 and NLRP3, together with the lack of melanoma 2 (AIM2) in gingival epithelial cells and macrophages of periodontitis patients, suggest their inflammatory role (*Park et al., 2014*). Therefore, many inflammatory molecules could be crucial in the development of early periodontal disease, including the presence of NO, the main inflammatory mediator in this disease (*Pacher, JS & Liaudet, 2007*).

It has been shown that high NO concentrations could be cytotoxic to periodontal tissue and related to probing depth and bone resorption (*Pansani et al., 2016*; *Wang, Huang & He, 2019*; *Reher et al., 2007*). In addition, it has been shown that NO has a side effect on the periodontal tissue, favoring vasodilation and platelet aggregation diminishes, which can contribute to gingival bleeding, aside from having cytotoxic effects on the surrounding tissue, increasing the severity of the periodontitis (*Boutrin et al., 2012*). The increase of NO levels in patients with periodontitis is related to high levels of iNOS expression in periodontal tissue cells during inflammation. Additionally, NO can function both as an apoptotic and an anti-apoptotic molecule, depending on its concentration and its interactions with other cellular molecules (*Sharma, Al-Omran & Parvathy, 2007*; *Shibata et al., 2001*).

On the other hand, ROS could be involved in the pathogenesis of many diseases, such as rheumatoid arthritis, chronic pulmonary disease, atherosclerosis, and recently in periodontitis (*Filippin et al., 2008*; *Soory, 2007*; *Tavakoli & Asmis, 2012*; *Waddington, Moseley & Embery, 2000*). Moreover, phagocytosed *P. gingivalis* by neutrophils is resistant to oxidative burst killing, partially by producing antioxidant enzymes such as superoxide dismutase, thiol peroxidase, and rubrerythrin (*Sztukowska et al., 2002*; *Mydel et al., 2006*; *Kikuchi et al., 2005*). Additionally, *P. gingivalis* accumulates a hemin layer on the cell surface that protects from oxidative stress (*Smalley, Birss & Silver, 2000a*; *Smalley, Birss & Silver, 2000b*). *P. gingivalis* can also survive within epithelial cells and induced the

production of ROS in the early stages of infection (*Choi et al., 2012*). Other studies suggest that ROS production induced by bacteria can activate JAK2 signaling (*Forget, Gregory & Olivier, 2005*) and increase levels of secreted IL-6 and IL-1 cytokines (*Wang et al., 2014*). However, it has also also been reported the advantages of using both mediators as cytotoxic molecules for *P. gingivalis.* The addition of nitric oxide and hydrogen peroxide has shown to kill *P. gingivalis*, (*Takada et al., 2017*). *Zou et al. (2022)* demonstrated that ROS production induces the viability loss of planktonic *P. gingivalis* and decreases its biofilm formation capacity.

*P. gingivalis* also can modulate cell death in various types of cells. LPS of *P. gingivalis* induces cell death in peripheral blood mononuclear cells (PBMCs) by early and late apoptosis, as well as necrosis through upregulation of Fas-FasL and activation of caspase-3 (*Trindade et al., 2012*; *Carvalho-Filho et al., 2019*). The gingipains of the bacterium promote apoptosis of human gingival fibroblasts through intracellular proteolytic activation of caspase-3 (*Stathopoulou et al., 2009*). Other cells, as endothelial cells (*Sheets et al., 2005*), cardiomyoblasts (*Lee et al., 2006*), lymphocytes (*Geatch et al., 1999*), monocytes (*Ozaki & Hanazawa, 2001*), and neutrophils (*Hiroi et al., 1998*; *Murray & Wilton, 2003*; *Preshaw & Taylor, 2011*), can be also induced to apoptosis by *P. gingivalis.* However, *P. gingivalis* also uses diverse mechanisms to modulate cell death pathways and ensure its intracellular survival. *P. gingivalis* promotes host-cell survival by inhibiting caspase-3 activation and balancing the expression of pro-apoptotic Bax and anti-apoptotic Bcl-2 (*Nakhjiri et al., 2001*; *Mao et al., 2007*). *P. gingivalis*, can replicate to high intracellular levels and does not induce host-cell death, in infected gingival epithelial cells (*Yilmaz et al., 2004*; *Mao et al., 2007*).

The oral antimicrobial peptides are the first line of innate defense in the oral. The antimicrobial capacity of saliva permits colonization of the oral cavity but prevents extensive microbial colonization of the oral and oropharyngeal tissues. The oral antimicrobial peptides also exert different immunomodulatory functions, including the regulation of the innate immune response (*Vray, Hartmann & Hoebeke, 2002*). Cystatin C is an antimicrobial peptide that decreases the growth of *P. gingivalis* by inhibiting proteolytic activity in the culture supernatant (*Blankenvoorde et al., 1998*). Some studies have shown that Ds-cystatin, a cystatin C homolog isolated from the tick *Dermacentor silvarum*, is internalized in LPS-stimulated mouse macrophages from *Borrelia burgdorferi,* inducing a decrease in inflammatory cytokines, such as IL-1β, IFN-γ, TNF-α, and IL-6 (*Sun et al., 2018*; *Gren et al., 2016*). In addition, cystatin C secreted by *Schistosoma japonicum* induces M2 macrophage polarization, favoring the production of anti-inflammatory cytokines such as IL-10 and TGF-β (*Yang et al., 2021*).

Due to the importance of controlling the inflammatory response during the development of periodontitis, in the present study, we investigated the effect of cystatin C on *P. gingivalis*-infected macrophages. We demonstrated that cystatin C has an important role in the intracellular antimicrobial activity and immunoregulatory activity on *P. gingivalis*-infected macrophages.

 

## MATERIALS & METHODS

### Isolation of peripheral blood mononuclear cells (PBMCs)

Peripheral blood mononuclear cells (PBMCs) were isolated from buffy coat cell packages donated for research and no longer usable for humans, obtained from the Blood Bank of the General Hospital of México "Eduardo Liceaga". PBMCs were isolated from blood by density gradient centrifugation using Histopaque-1077 (Sigma-Aldrich, St. Louis, MO, USA) at $325 \times$ g for 20 min at 20 °C. Briefly, after 1:1 dilution with PBS pH 7.2 (Gibco Life Technologies, Waltham, MA, USA), blood samples are slowly and steadily placed on top of Histopaque-1077 (Sigma-Aldrich), in centrifuge tubes and centrifuged at $325 \times$ g for 10 min. The cells were carefully harvested from the interphase, washed with cold PBS pH 7.2 buffer, and incubated for 15 min in lytic solution to lyse residual erythrocytes. The PBMCs obtained were washed with PBS pH 7.2 and counted in a hemocytometer (Millipore, Burlington, MA, USA) using the trypan blue dye exclusion method.

The study was performed in accordance with the Declaration of Helsinki and approved by the Ethics Committee of the Universidad Nacional Autónoma de México (reference number FM/DI/038/2021).

### CD14$^+$ monocytes isolation and macrophages differentiation

Monocytes were isolated from previously obtained PMBC by positive selection with anti-human CD14 microbeads (Miltenyi Biotech, Bergisch Gladbach, Germany), according to the manufacturer's protocol.

Human CD14$^+$ monocytes were seeded in low adherence 24-well plates (Costar 3473) with a cell density of $1 \times 10^6$/well, in RPMI-1640 medium (Sigma-Aldrich, USA, R6504-1L) supplemented with 10%, 2 mM L-glutamine, 100 U/ml penicillin, 0.1 mg/mL streptomycin and 10% fetal bovine serum (FBS) (GIBCO BRL, Gaithersburg, MD, USA) at 37 °C in a humidified 5% $CO_2$ incubator for 7 days.

The adherent macrophages were then detached *via* cold EDTA solution at 0.02% (Sigma-Aldrich, USA), collected in 50 ml tubes, and centrifuged at $325 \times$ g for 5 min at 4 °C. The cells were resuspended in RPMI-1640 (R6504-1L; Sigma-Aldrich) supplemented with 10%, 2 mM L-glutamine, 100 U/mL penicillin, and 0.1 mg/mL streptomycin and 10% FBS (GIBCO BRL, Gaithersburg, MD, USA). The recovered macrophages were counted with a hemocytometer (Millipore, Burlington, MA, USA) using the trypan blue dye exclusion method.

### Monocyte-derived macrophage purity evaluated by flow cytometry

Monocytes-derived macrophages purity was evaluated using flow cytometry (FACScan). Monocytes were resuspended in 100 μL buffer, pH 7.2 and stained with 2 μL anti-human CD14-PE (1:50) (BD Pharmigen, San Diego, CA, USA [clone M5E2]) for 30 min in the dark at 4 °C.

Differentiated macrophages were resuspended in 100 μL of buffer, pH 7.2, and stained with 2 μL of anti-human CD68-PE (1:50) (BD Pharmigen [clone M5E2]) for 30 min in the dark at 4 °C. Analysis was performed with a Flow Jo 10 software (BD Biosciences, Heidelberg, Germany).

## Bacterial growth

*P. gingivalis* strain ATCC 33277 was cultured as previously described (*Blancas-Luciano et al., 2022*). Briefly *P. gingivalis* was cultured in brain-heart-infusion (BHI; BD Bioxon, Milan, Italy) containing 5 mg/mL of hemin (Sigma-Aldrich, Munich, Germany) and 1 mg/mL of menadione (Sigma-Aldrich) under anaerobiosis using the anaerobic BBL-GasPak jar system (BD Biosciences), at 37 °C for 24 h.

After 24 h of culturing, bacteria were harvested by centrifugation for 10 min at 8,200 x $g$ and then washed and resuspended in Krebs-Ringer-Glucose (KRG) buffer (120 mM NaCl, 4.9 mM KCl, 1.2 mM MgSO$_4$, 1.7 mM KH$_2$PO$_4$, 8.3 mM Na$_2$HPO$_4$, 10 mM glucose, and 1.1 mM CaCl$_2$, pH 7.3). Bacterial growth was monitored spectrophotometrically (Jenway Genova R0027; Thermo Fischer Scientific, Waltham, MA, USA) at 675 nm. The bacterial density was visually adjusted to a turbidity of 0.5 McFarland ($1 \times 10^8$ colony-forming units); (CFU/mL) (*Bollela, Sato & Fonseca, 1999*; *Emani, Gunjiganur & Mehta, 2014*).

## Cystatin C

We used human recombinant Cystatin C expressed in *Pichia pastoris* (Sigma Aldrich, St. Louis, MO). It was reconstituted with one mL of buffer Tris-Base NaCl (pH 7.4).

## Cystatin C antimicrobial activity assays against *P. gingivalis*

The antimicrobial activity of cystatin C was evaluated by microdilution technique (*Eloff, 1998*; *Jadaun et al., 2007*) using the commercially available growth indicator Presto Blue. Briefly, $5 \times 10^5$ CFU/mL of *P. gingivalis* per well were seeded in KRG buffer in 96-well plates (Costar; Corning Life Sciences, Corning, NY, USA). Cystatin C (1.25, 1.5, 1.75, 2, 2.25, 2.5, 2.75 µg/mL) was incubated for 1, 12, 24, 48, and 96 h at 37 °C under anaerobic conditions. After the incubation period, 20 µL of Presto Blue Cell Viability Reagent (Invitrogen, Thermo Fisher Scientific) per well was added. The plates were incubated for 30 min at 37 °C in the dark. Finally, the plates were read in a microplate reader (Multiskan SkyHigh Microplate Spectrophotometer; Thermo Fisher Scientific), at a 675 nm wavelength.

## Cell viability assay

Macrophages were seeded in a 96-well (Costar, Corning Life Sciences) plate at a density of $1 \times 10^5$ cells/well and incubated for 12 h at 37 °C with 5% CO$_2$. Next, different concentrations of cystatin C (1.25, 1.5, 1.75, 2, 2.25, 2.5, 2.75 µg/mL) were added and incubated for 24 h. At the end of the incubation, 25 µl/well of XTT/PBS solution (4 mg/mL) for 40 min at room temperature in the dark, was added. The samples were read at a wavelength of 450 nm in a microplate spectrophotometer (Multiskan SkyHigh Microplate Spectrophotometer). The method was first used in the methods section from *Blancas-Luciano et al. (2022)*.

## Intracellular survival and invasion assay

Macrophages were seeded in Costar® 24-well plate (Corning Life Sciences, Corning, NY, USA) at a cell density of $1 \times 10^6$/well in RPMI-1640 medium (R6504-1L; Sigma-Aldrich) supplemented with 10%, 2 mM L-glutamine, for 12 h at 37 °C and 5% CO$_2$.

At the end of the incubation, the medium was removed and replaced with fresh medium without antibiotics. Cells were infected with *P. gingivalis* at MOI: 1:10 for 1, 3, 24, 48, and 96 h. The infected macrophages were stimulated with cystatin C for 24 h. Subsequently, the cells were washed three times with PBS and harvested using 0.02% EDTA solution (Sigma-Aldrich).

Invasion of *P. gingivalis* was quantitated by a standard antibiotic protection assay as described previously (*Lamont et al., 1995*; *Yilmaz, Watanabe & Lamont, 2002*). Briefly, infected macrophages and stimulated with cystatin C and washed twice and treated with metronidazole (200 μg/ml) and gentamicin (300 μg/ml) for 1 h to eliminate remaining extracellular bacteria. After exposure to the antibiotic, cells were washed twice with PBS pH 7.2 and lysed in one mL of cold sterile distilled water per well for 30 min. Cell lysates were serially diluted and plated on blood agar plates (BHI, Becton Dickinson, Sparks, MD, USA) supplemented with hemin and menadione, and incubated anaerobically at 37 °C for 7 days. Colony forming units per mL were then enumerated (Supplementary figure).

## Antibody production

Antisera against cystatin C was generated in female CD-1 mice weighing $20 \pm 2$ g. The mice were immunized for 6 weeks at 8-day intervals with 100 μg/100 μL Cystatin C. The immunizations were administered intraperitoneally for 4 weeks and subcutaneously for 2 week. For the first intraperitoneal immunization, Freund's complete adjuvant was added, and for the second intraperitoneal immunization, Freund's incomplete adjuvant was included. Five weeks after the first immunization, mice were anesthetized to obtain whole blood, which was aliquoted and stored at $-80$ °C until use. Antibody titers were estimated by an enzyme-linked immunosorbent assay, according to *Fernández-Presas et al. (2018)*. Guidelines for laboratory animal care established by the ethical committee of the School of Medicine, Universidad Nacional Autónoma de México were strictly followed.

## Confocal microscopy

Cystatin C-stimulated *P. gingivalis*-infected macrophages were incubated in 24-well plates containing glass coverslips in RPMI-1640 (R6504-1L; Sigma-Aldrich, USA) supplemented with 10% FBS (GIBCO BRL, Gaithersburg, MD, USA). Briefly, cells attached to coverslips were washed with PBS, fixed with 2% paraformaldehyde (PFA) for 20 min at 4 °C, and washed with 1x Perm/Wash buffer (Biolegend). Staining was done with mouse anti-Cystatin C primary antibodies diluted 1:25 in permeabilization solution for 30 min at 4 °C. Cells were washed again with 1x Perm/Wash. Subsequently, it was stained with mouse IgG secondary antibody (Cy3) diluted 1:25 in permeabilization solution for 30 min at 4 °C. Cell nuclei were counterstained with DAPI (1:1000) for 3 min in fluoroshield mounting medium (Sigma-Aldrich, USA). The cells from experimental groups and controls are visualized by confocal microscopy (Leica, TCS SP5; Leica Microsystems, USA), using LAS X v software 3 for image analysis.

## Cytokine assays

Macrophages were infected with *P. gingivalis* and stimulated with cystatin C for 24 h at 37 °C with 5% $CO_2$. The cytokines IL-1β, TNF-α and IL-10 were quantified by ELISA in the cell supernatants. Human cytokine ELISA kits (BD Biosciences Cytokine ELISA Protocol) were used according to the manufacturer's instructions. Briefly, anti-TNF-α, an-ti-IL-1β, and anti-IL-10 capture antibodies (BD Bioscience, Pharmingen) were placed in 96-well flat-bottom plates (Costar; Corning Life Sciences, Corning, NY, USA) diluted in 0.1 M $Na_2HPO_4$, pH 6, and incubated overnight at 4 °C. The wells were blocked with PBS solution supplemented with 0.5% casein diluted in 0.1 N NaOH. Cell supernatants and those from the TNF-α standard curve, recombinant IL-1β, and IL-10 (BD Bioscience, Pharmingen) were incubated in a D-MEM culture medium overnight at 4 °C. To detect TNF-α, IL-1β, and IL-10, the detection antibodies anti-TNF-α, anti-IL-1β, and anti-IL-10 biotinylated (BD Bioscience) were diluted in 1% BSA with 0.05% Tween 20 and incubated at room temperature for 1 h. The plate was analyzed using a streptavidin-alkaline phosphatase (AP) conjugate (Thermo Fisher Scientific, Life Technologies) and 0.005 mg/mL phosphatase substrate (Sigma Aldrich). Control groups included macrophages without stimulation or stimulated with LPS (100 ng/mL) (LPS from *Escherichia coli* O111:B4; Sigma Aldrich) or with a peptidoglycan (10 μg/mL) (peptidoglycan from *Staphylococcus aureus*, (Sigma-Aldrich). Absorbance was quantified in a spectrophotometer (Multiskan SkyHigh Microplate Spectrophotometer) at 405 nm using Ascent Multiskan V. 2.6 software. Cytokine concentrations in samples were determined from the standard curve by regression analysis.

## Nitric oxide production

Macrophages infected with *P. gingivalis* were stimulated with 2.75 μg/mL cystatin C for 48 h at 37 °C with 5% $CO_2$. Subsequently, the supernatants were isolated, and their nitrite concentration was quantified using the Griess Reaction kit (Life Technologies, Carlsbad, CA, USA), according to the manufacturer's instructions. Supernatants from macrophages infected with *P. gingivalis* and stimulated with cystatin C were placed in a 96-well microplate and 20 μL Griess reagent (1% sulfanilamide in 5% phosphoric acid and 0.1% naphthyl ethylenediamine-HCl) were added (Sigma Aldrich). The cells were incubated at room temperature for 30 min. The absorbance was measured at 550 nm with a microplate reader (Multiskan SkyHigh Microplate Spectrophotometer). The culture medium was used as a blank in all experiments. The number of nitrites in the test samples was calculated from a standard curve of sodium nitrite.

## Production of reactive oxygen species

Macrophages infected with *P. gingivalis* and stimulated with cystatin C (2.75 μg/mL) were incubated for 24 h at 37 °C with 5% CO2. Subsequently, the cells were incubated with 2′,7′-dichlorodihydrofluorescein diacetate ($H_2$DCFDA) at a concentration of 100 μM for 30 min. Cells were washed with PBS pH 7.2 and detached from plates with 0.02% EDTA solution (Sigma Aldrich, St. Louis, MO, USA). The samples were resuspended in PBS pH 7.2 with 1% FBS and analyzed in a BD bioscience FACS Canto II flow cytometer.

Data analysis was performed using FlowJo 10 software (https://www.flowjo.com/solutions/flowjo/downloads).

## Transmission electron microscopy (TEM)

The cell pellets of infected macrophages and stimulated with cystatin C were processed for standard transmission electron microscopy as previously described (*Fernández-Presas et al., 2018*). Briefly, all samples were fixed in a mixture of 1.5% glutaraldehyde and 4% paraformaldehyde, buffered in PBS (pH 7.2), at room temperature for 1 h. The samples were postfixed with 1% osmium tetraoxide for 4 h. All samples were subsequently dehydrated by incubation in a series of ascending concentrations of ethanol and embedded in an epoxy resin (Poly/Bed812/DMP30; Polysciences, Warrington, PA, USA).

Ultrathin sections (50–60 nm) were mounted on formvar-coated copper grids, stained, using the double-contrast technique, with uranyl acetate for 30 min followed by lead citrate for 15 min. Later, the sections were rinsed with deionized water and dried. Subsequently, they were examined at 80 kV with a JEOL l JEM 1200EXII transmission electron microscope (JEOL, Peabody, MA, USA). Depending on the structure of interest, multiple microphotographs were taken with a magnification range of $5,000\times$ to $1,20\ 000\times$.

## TUNEL assay

A terminal deoxynucleotidyl transferase dUTP nick end labeling (TUNEL) assay (*in situ* cell death detection kit; Boehringer Mannheim; Ingelheim am Rhein, Germany) was used for the detection of cell death by apoptosis using enzymatic labeling of the DNA strand breaks with fluorescein dUTP and TdT (*Gorczyca, Gong & Darzynkiewicz, 1993*).

*P. gingivalis*-infected macrophages stimulated with cystatin C were washed twice in PBS/1% bovine serum albumin and subsequently resuspended in paraformaldehyde (2%) in PBS (pH 7.4) and incubated for 30 min at room temperature. Cells were washed three times (centrifuged at 1,300 g) for 10 min and suspended in permeabilization solution (0.1% Triton X-100 in 0.1% sodium citrate) for 2 min on ice. The cells were washed twice in PBS at pH 7.4 and 50 µL TUNEL reaction mix was added. The cells were incubated for 60 min at 37 °C in a humid chamber. Macrophages stimulated with DNase I (Sigma; 600 U/ml in 50 mM Tris–HCl, pH 7.5, 1 mg/ml BSA) for 10 min were used as a positive control. The samples were washed twice in PBS and analyzed by flow cytometry (*Fernández-Presas et al., 2001*).

## Annexin-V

Exposure of phosphatidylserine (PS) on the cell surface was analyzed by annexin-V binding and cell membrane integrity by 7-AAD staining, using the Annex-in-V-FLUOS Staining Kit (Roche Diagnostics GmbH, Mannheim, Germany) following the protocol recommended by the manufacturer. Briefly, macrophages infected with *P. gingivalis* ($1.5\times 10^5$) and stimulated with cystatin C (2.75 µg/mL) were washed twice in PBS, pH 7.4, and subsequently suspended in staining solution (Annexin-V-Fluos/Hepes buffer [10 mM Hepes/NaOH, pH 7.4, 140 mM NaCl, 5 mM $CaCl_2$], supplemented with propidium iodide [50 µg/mL]) for 15 min at 25 °C in the dark. Cells incubated with 4 µg/mL of

camptothecin (Roche) for 3 h, were used as a positive control. Samples were analyzed by flow cytometry on a FACSort with CellQuest software (BD Immunocytometry Systems, San Jose, CA, USA). Data were analyzed using the Flow Jo software (Becton, Dickinson, Milpitas, CA, USA). The results were presented as the percentage of viable (Ann⁻V⁻PI⁻), early apoptotic (Ann⁻V⁺PI⁻), or nonviable (Ann⁻V⁻PI⁺) cells.

## Caspase 3 activity

The assay was performed as recommended by the manufacturer. Briefly, cells were washed twice with cold PBS and homogenized in standard buffer [100 mM Hepes; 10% (w/v) sucrose 0.1% w/v) CHAPS; 10 mM DTT; 1 mM EDTA; 1 mM PMSF; 1 mM Naf; 2 μg/mL aprotinin; 1 μg/mL pepstatin; and 5 μg/mL leupeptin]. Macrophages infected with *P. gingivalis* and stimulated with cystatin C were transferred to well plates at a density of $1 \times 10^4$ cells/well. The reaction buffer supplemented with the caspase 3 substrate MOCAc-DEVDAPKb (Dnp)-NH2 (25 μM) was incubated for 4 h at 37 °C. Caspase 3 activity was measured by the release of p-nitroaniline (pNA). The p-nitroaniline was detected at 405 nm with a luminescence spectrometer (Fluoroskan Ascent Thermo Labsystems, Helsinki, Finland). The concentration of pNA released from the substrate was calculated us-ing a calibration curve prepared with pNA standards (pNA standard included in the kit). This method was first used in the Methods section from *Fernández-Presas et al. (2010)*.

## Statistical analysis

Experimental and control conditions were statistically compared for significance using analysis of variance (ANOVA), followed by Benferroni correction. The predetermined level of significance was $p < 0.05$. Statistical analysis was performed with the GraphPad, Prism v.6 software (GraphPad Software, Inc., La Jolla, CA, USA).

# RESULTS

## Cystatin C inhibits extracellular growth of *P. gingivalis* without altering macrophage viability

To analyze the extracellular antimicrobial activity of cystatin C against *P. gingivalis*, bacteria were incubated with different concentrations of the peptide (1.25, 1.5, 1.75, 2, 2.5, 2.75 μg/mL) and increasing time periods (1, 12, 24, 48, and 96 h). Cystatin C exhibited a dose-and time-dependent antimicrobial activity after 12 h of incubation, as shown in Fig. 1A. Additionally, maximum microbicidal activity was exhibited at concentrations between 2.25 and 2.75 μg/mL after 12 h. Concentrations of 2.25 μg/mL, 2.5 μg/mL, and 2.75 μg/mL inhibited bacterial growth at 60%, 80%, and 95%, respectively, after 24 h of incubation and compared to the control group ($p < 0.05$). Furthermore, bacterial inhibition with these three concentrations was also observed after 48 h of incubation. Interestingly, a strong bacterial growth inhibition was observed at a concentration of 2.75 μg/mL during all the incubation times (Fig. 1A). Therefore, we decided to perform all the experiments with this concentration. No significant differences in the cell viability were observed after 24 h when macrophages were incubated with different concentrations of cystatin C (Fig. 1B).

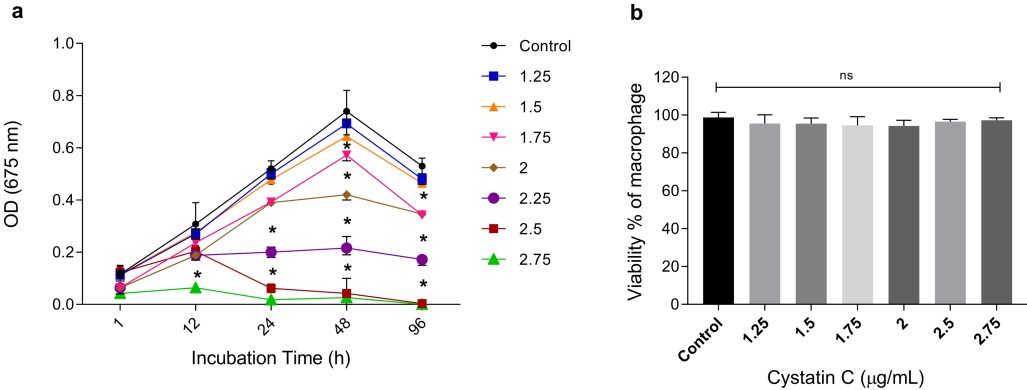

**Figure 1** **Effect of cystatin C on *P. gingivalis* growth and macrophage viability.** (A) $5 \times 10^6$ CFU/mL of *P. gingivalis* were cultured in the presence of cystatin C at different concentrations, for 1, 12, 24, 48, and 96 h. Cells without stimulation were used as a control group. when compared to the control group. The significance represents a $p < 0.05$ (*). (B) $1 \times 10^5$ cells were co-incubatedat different concentrations of cystatin C for 24 h. No significant differences were found when comparing different concentrations of cystatin C with control group p& γ τ 0.05 (ns). Data represent the mean ± SD of five independent experiments ($n = 5$).

## Cystatin C is internalized and exhibits intracellular antimicrobial activity in macrophages infected with *P. gingivalis*

Different assays were performed to demonstrate that *P. gingivalis* is internalized in the macrophage and that cystatin C participates in reducing the intracellular bacterial load. Survival of the bacterium inside a macrophage was determined using the CFU count and performing an invasion assay. Macrophages were infected at (MOI 1:10) and *P. gingivalis* was internalized during the first 3 h of infection, then macrophages were exposed to cystatin C. In 88.54% of the cells and survived inside the macrophage during the next 48 h (Fig. 2). In contrast, cystatin C inhibited the intracellular bacterial load 37.5% at 24 h and 79% at 48 h after macrophage stimulation. Additionally, the incubation of macrophages with bacteria and with the peptide for 96 h eliminated the bacterium in 97% (Fig. 2). These findings demonstrate that the stimulation of macrophages with the peptide decreased the intracellular bacterial load of *P. gingivalis*.

Additionally, immunofluorescence analysis was performed to localize cystatin C in infected macrophages. CFDA-labeled *P. gingivalis* was in the cytoplasm of the cell (red arrows) (Fig. 3), whereas cystatin C was located in the cytoplasm and the plasma membrane of basal macrophages (white arrows) (Fig. 3), confirming the internalization capacity exhibited by the peptide in these cells. In the infected macrophages, a co-localization of *P. gingivalis* and cystatin C was observed, which suggests that this peptide could be acting directly on the bacteria to reduce its intracellular survival.

## Cystatin C decreases the production of inflammatory cytokines and increases the production of anti-inflammatory cytokines

TNF-α and IL-1β were evaluated in supernatants of macrophages (MOI 1:10) infected with *P. gingivalis* and after 3 h of the infection with the bacterium , macrophages were

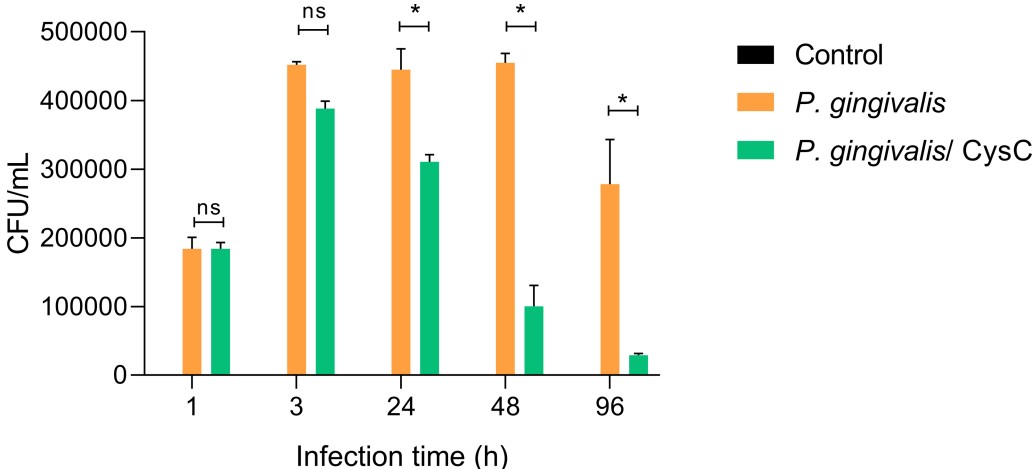

**Figure 2** *P. gingivalis* **intracellular survival assay.** Macrophages were infected with *P. gingivalis* (MOI: 1:10) for 1, 3, 24, 48, and 96 h.Infected macrophages were stimulated with cystatin C for 24 h and treated with metronidazole (200 µg/mL) and gentamicin (300 µg/mL) to eliminate all extracellular bacteria. Macrophages were lysed at the indicated times and lysates were plated on anaerobic blood agar plates to determine CFU/mL. Macrophages lysates without bacterium and without stimulation were used as a control groups. Statistical differences were observed between macrophages infected with *P. gingivalis*, and macrophages infected with *P. gingivalis* plus cystatin C after 24, 48 and 96 h ($p < 0.05$) and are represented with asterisks (*). No statistical differences were observed after 48 and 96 h ($p > 0.05$). Data represent the mean ± SD of five independent experiments ($n = 5$).

exposed to cystatin C (2.75 µg/mL) for 24 h. *P. gingivalis* induced the production of 1,500 pg/mL and 950 pg/mL of TNF-α and IL-1β, respectively, in the macrophage (Figs. 4A, 4B). However, stimulation of the macrophage with cystatin C significantly reduced TNF-α (500 pg/mL) and IL1β (500 pg/mL) ($p < 0.05$) production, as compared to infected macrophages (Figs. 4A, 4B).

Additionally, we analyzed the production of IL-10 in macrophages infected with *P. gingivalis*. As shown in Fig. 4C, *P. gingivalis* did not induce the production of IL-10 in macrophages. Interestingly, macrophages incubated with cystatin C increased the production of this cytokine (760 pg/mL). In addition, this increase was observed in macro-phages infected with *P. gingivalis* and stimulated with cystatin C (1,100 pg/mL), compared to the control group (Fig. 4C).

These results suggest that cystatin C contributes to the regulation of the inflammatory process, reducing the production of inflammatory cytokines and increasing the production of anti-inflammatory cytokines.

## Cystatin C decreases NO production in macrophages and increases ROS production in macrophages infected with *P.gingivalis*

ROS and NO are inflammatory mediators induced in response to *P. gingivalis* infection. The bacterium increased the concentration of ROS and decreased NO production 4 and 9 times, respectively, when compared to the uninfected control group (Figs. 5A, 5B). Surprisingly, stimulation of macrophages with cystatin C also induced ROS production, 2-fold compared to basal macrophages. Furthermore, cystatin C also increased ROS

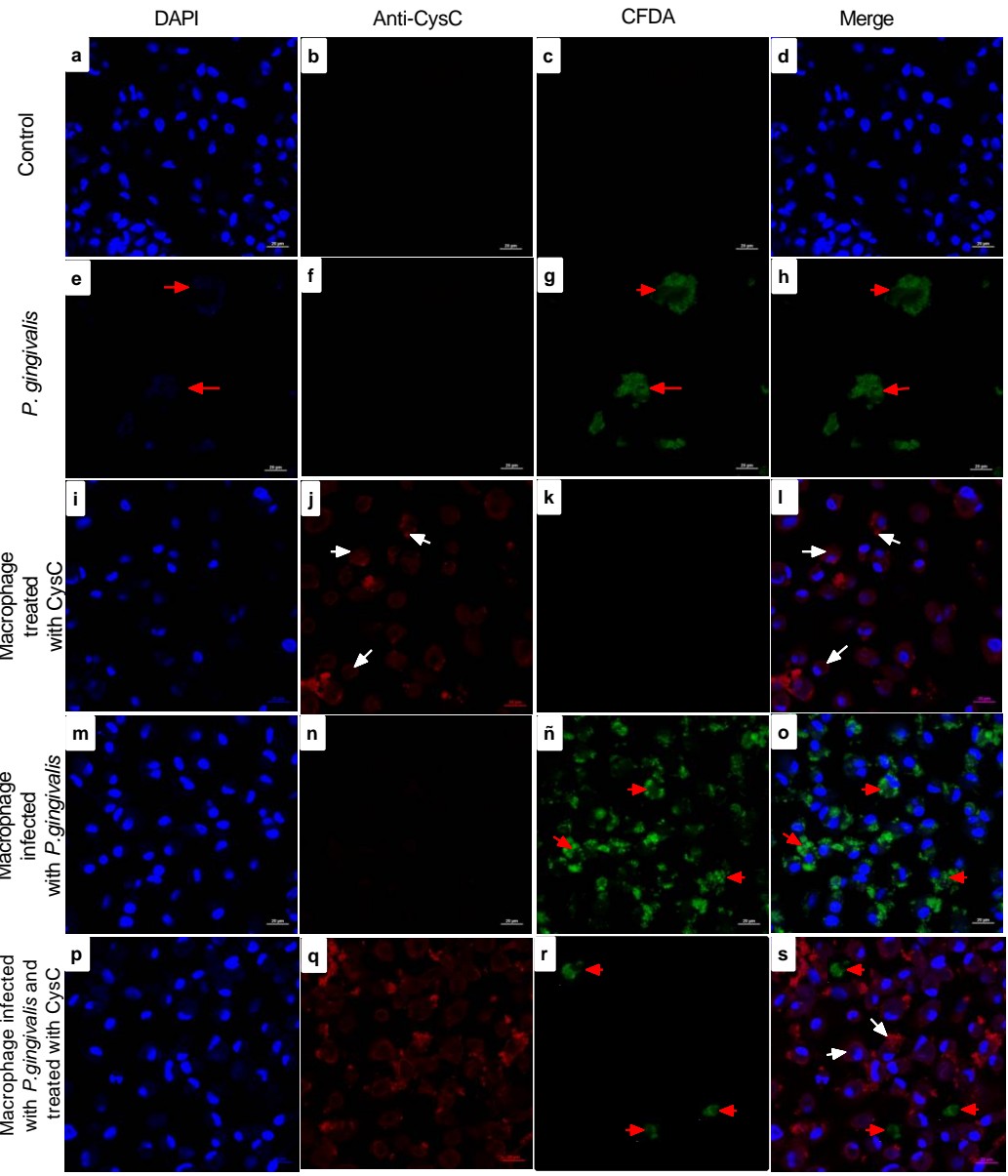

**Figure 3 Localization of cystatin c in macrophages infected with *P. gingivalis* and stimulated with cystatin C for 24 hours.** Macrophages were stained with: DAPI(blue); (A), (I), (M), (P), anti-cystatin C (Anti Cy3- bright red) (B), (F), (J), (N), (Q). *P. gingivalis* was stained with CFDA (green) (C), (G), (K), (ñ), (R). Merge (D), (H), (L), (O), (S). Red arrows exhibit: Intracellular bacteria in the macrophage. White arrows show localization of cystatin C in the macrophage. Scale bar 20 μm.

production 2-fold in macrophages infected with *P. gingivalis*, compared to the group of infected macrophages (Fig. 5A).

NO production in the macrophage infected with *P. gingivalis* was significantly increased (9 μM) compared to the control group. Yet, incubation of macrophages with the bacterium and cystatin C leads to a decrease in NO production (3 μM) (Fig. 5B).
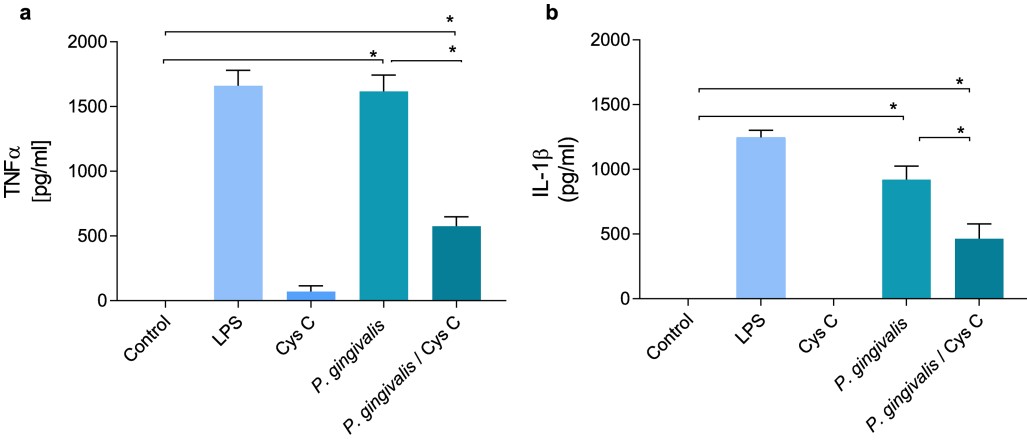

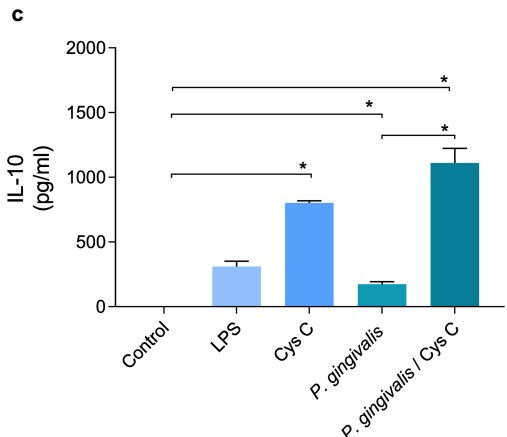

**Figure 4** **Cytokine production in macrophages infected with *P. gingivalis* and stimulated with cystatin C.** Macrophages were infected with *P. gingivalis* (MOI 1:10) for 3 h and stimulated with cystatin C for 24 h. Basal and LPS-stimulated cells were used as controls. Supernatants from experimental and control groups were analyzed by ELISA. (A) TNF-α expression, (B) IL-1β expression, (C) IL-10 expression. Statistical differences in the expression of TNF-α and IL-1β comparing control group with *P. gingivalis* $p <$ 0.05. Statistically significant were also observed when comparing control group with *P. gingivalis* plus cystatin C ($p < 0.05$). Significant differences comparing *P. gingivalis* with *P. gingivalis* plus cystatin C were observed ($p < 0.05$). No statistical differences were observed when comparing control group with cystatin C ($p\& γ τ$ ;0.05). Statistical differences in the expression of IL-10 were observed when comparing control group with cystatin C ($p < 0.05$), and when comparing control group with *P. gingivalis* plus cystatin C ($p < 0.05$). Furthermore, significant differences were observed when comparing *P. gingivalis* with *P. gingivalis* plus Cystatin C ($p < 0.05$). The statistical difference are represented with asterisks (*). Data represent the mean ± SD of five independent experiments ($n = 5$).

The obtained results showed that cystatin C could contribute to the inflammatory mediator's reduction such as NO, which can favor the regulation of tissue damage induced during infection with *P. gingivalis*. At the same time, cystatin C increases ROS production, which may be related to the antimicrobial activity observed in Fig. 1A.

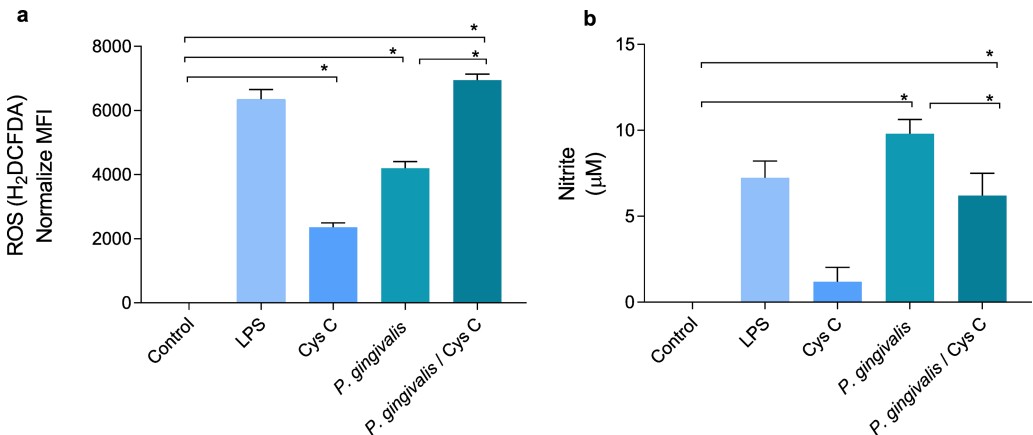

**Figure 5** **Production of reactive oxygen species and nitric oxide in macrophages infected with *P. gingivalis* and stimulated with cystatin C.** Macrophages were stimulated with *P. gingivalis* (MOI 1:10) for 3 h and stimulated with cystatin C for 24 h. Basal cells and cells stimulated with LPS were used as controls. (A) ROS-production for flow cytometry in experimental and control groups. Statistical differences in the production of ROS were observed when comparing control group with cystatin C and between *Porphyromonas gingivalis* and with *P. gingivalis* plus cystatin C ($p < 0.05$). Significant differences were also observed when comparing *P. gingivalis*, with *P. gingivalis* plus cystatin C ($p < 0.05$). (B) Expression of nitrites in supernatants of experimental groups and controls incubated for 48 h. Statistical differences were observed in the NO production when comparing control group with *P. gingivalis*, and with *P. gingivalis* plus cystatin C ($p < 0.05$). Significant differences were also observed when comparing *P. gingivalis* with *P. gingivalis* plus cystatin C ($p < 0.05$). The statistical difference are represented with asterisks (*). No statistical differences were observed when comparing control group with cystatin C ($p < 0.05$). Data represent the mean ± SD of five independent experiments ($n = 5$).

## Macrophages infected with *P. gingivalis* and with cystatin C exhibit a normal cell shape

The ultrastructure of macrophages incubated with *P. gingivalis* was analyzed by TEM (Fig. 6). Macrophages without treatment showed a homogeneous electron-density in the cytoplasm and nucleus, as well as intact plasma and nuclear membranes (Fig. 6A). After infection, the bacterium is depicted as a small electrodense particle distributed in the cytoplasm of the macrophage after 24 h of incubation, where no morphological changes are evident in macrophages infected with bacteria after 24 h of incubation (Fig. 6B). Yet, after 96 h of incubation, macrophages infected with bacteria exhibited extensive damage, showing plasma- and nuclear membrane disruptions, chromatin attachment to the nuclear membrane, and vacuole leakage of cell contents (Fig. 6D). In contrast, macrophages infected with *P. gingivalis* and co-incubated with cystatin C showed intact plasma and nuclear membranes, and a homogeneous electron density of the cytoplasm and nucleus (Figs. 6E–6G). Figure 6H shows the bacterium entering the macrophages. These results suggest that cystatin C could prevent the damage induced by the bacteria.

## Cystatin C decreases cell apoptosis in macrophages infected with *P. gingivalis*

After, the observed findings in the transmission electron micrographs, our next objective was to identify the cell death induced by *P. gingivalis* and the effect of cystatin C on

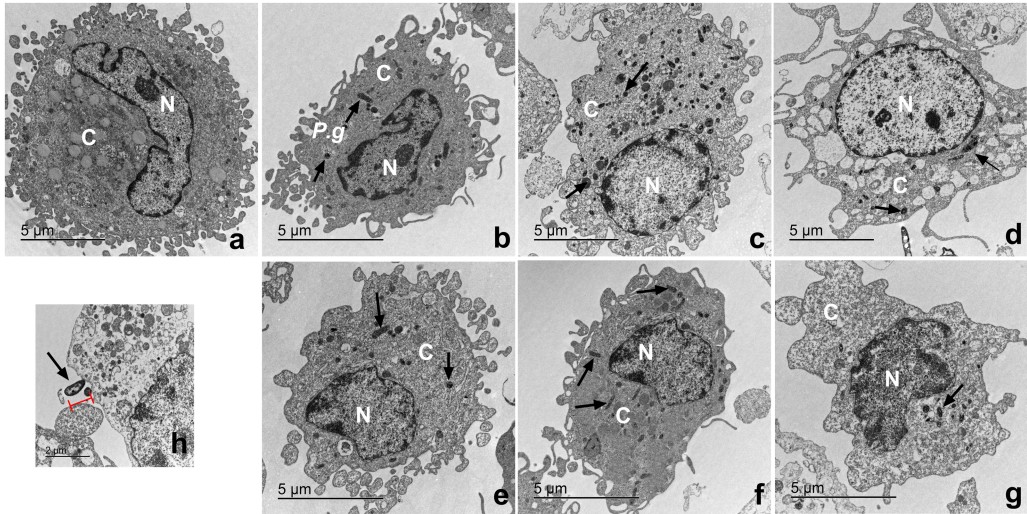

**Figure 6** **Transmission electron microscopy (TEM) of macrophages infected with *P. gingivalis* and treated with cystatin C.** Cell pellets of cystatin C-stimulated infected macrophage for standard transmission electron microscopy were processed. (A) Basal macrophages. Macrophages infected with *P. gingivalis* during (B) 24 h, (C) 48 h, (D) 96 h. Macrophages infected with *P. gingivalis* and stimulated with cystatin C for (E) 24 h, (F) 48, (G) 96 h. Black arrows indicates the presence of *P. gingivalis*. (H) Note bacterium entering the macrophages (arrow).

this phenomenon. For this, we carried out three tests; TUNEL, Annexin V activity, and caspase 3.

## Detection of DNA strand breaks

The TUNEL assay was performed to detect apoptotic cell death by enzymatic labeling of DNA strand breaks with fluorescein dUTP and TdT. The control groups were, macrophages non-stimulated (Fig. 7A), and incubated with DNase 1 (Fig. 7B). Incubation with cystatin C-stimulated macrophages (Fig. 7C) did not exhibit cell apoptosis. In contrast, cells incubated with *P. gingivalis* for 96 h showed 57% macrophage apoptosis compared to the uninfected control group (Figs. 7D, 7F). Interestingly, cystatin C significantly decreased cell apoptosis by 20%, in macrophages infected with *P. gingivalis* (Figs. 7E, 7F).

## Detection of phosphatidylserine on the surface of macrophages

Additionally, the cell death of macrophages was studied by the surface expression of PS using Ann-V in conjunction with propidium iodide (PI) and fluorescein-activated cell sorting (FACS) analysis. Untreated cells were used as a negative control (Fig. 7G) and DNase I-treated cells as a positive control (Fig. 7H). The percentage of apoptotic cells in the infected macrophage group was 78.3% (Figs. 7J, 7L). After macrophage simulation with cystatin C, (Fig. 7I), the percentage of apoptosis in infected macrophages decreased to 30.5%, as shown in Figs. 7K, 7L.

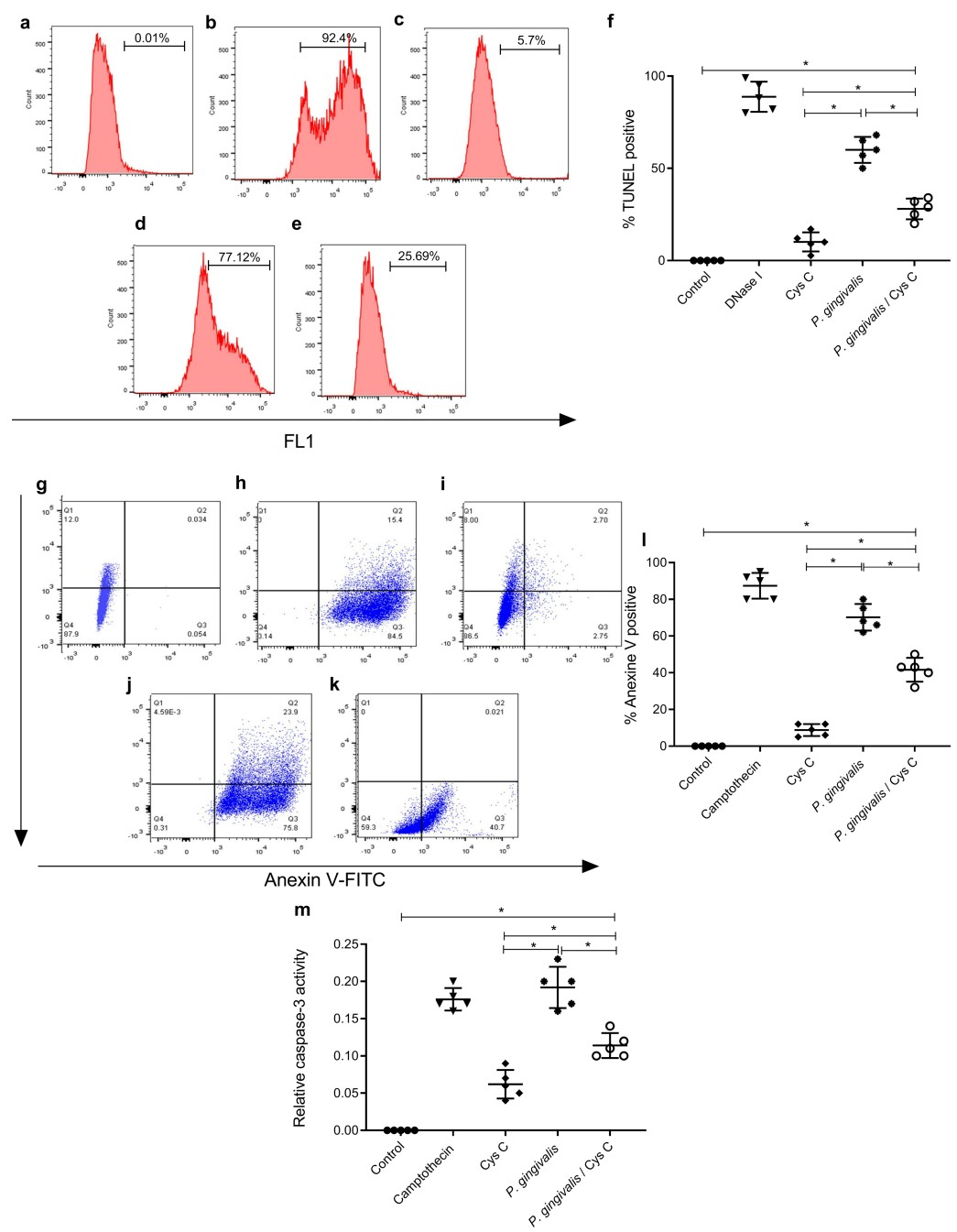

**Figure 7   Effect of cystatin C on apoptotic activation of macrophages infected with *P. gingivalis*.**
TUNEL staining scatter plots after 96 h of incubation of (A) basal macrophages, (B) macrophages
incubated with DNase I (600U/mL), (C) macrophages stimulated with cystatin C, (D) macrophages
infected with *P. gingivalis*, (E) macrophages infected with P. gingivalis and stimulated with cystatin C. (F)
Percentage of TUNEL positive cells. Cells without stimulation and stimulated with DNase I as a control
group, were used. Scatter plots of Annexin V staining after 96 h incubation of (G) basal macrophages,

**Figure 7 (…continued)**
(H) camptothecin (4 μg/mL) incubated macrophages, (I) cystatin C-stimulated macrophages, (J) *P. gingivalis*-infected macrophages, (K) macrophages infected with *P. gingivalis* and stimulated with cystatin C. (Annexin-V2PI2 [viable cells], Annexin V+PI+ [late or necrotic apoptosis), Annexin V+PI3 [cellular apoptosis]. (L) Percentage of cells Annexin V positive cell. Cells without stimulation and stimulated with camptothecin are used as control group. (M) Activity of Caspase-3 on macrophages infected with *P. gingivalis*. Macrophages were infected with *P. gingivalis* (MOI 1:10) for 96 h and stimulated with cystatin C for 24 h. Cells without stimulation and stimulated with camptothecin are used as control group. Data represent the mean ± SD of five independent experiments ($n = 5$). The significance represents a $p < 0.05$ (*) with respect to the control group.

## Caspase-like proteases are involved in macrophage death

The morphological and cellular changes associated with apoptosis are due to the activation of the caspase cascade, most notably the executioner caspase 3 (*Earnshaw, Martins & Kaufmann, 1999*). Caspase-3 activation plays a key role in the initiation of cellular events during the early apoptotic process. We examined whether the bacterium inside macrophages can induce caspase-3 activation and the effect of cystatin C in this activation.

Our results showed a significant increase in macrophage caspase-3 activity after 96 h of challenging with *P. gingivalis*, compared to the control group (Fig. 7M). Yet after adding Cystatin C, a significant decrease in macrophage caspase-3 activity was observed (Fig. 7M).

As a whole, these results show that macrophage apoptosis occurs after exposure to *P. gingivalis* and that cystatin C could participate by inhibiting caspase 3 activity, thereby, decreasing the apoptotic process observed in macrophages infected with *P.gingivalis*.

## DISCUSSION

In this study, we sought to determine whether cystatin C reduces the intracellular growth of *P. gingivalis* in infected macrophages and to evaluate the inflammatory response induced by this bacterium in macrophages. Our results show that cystatin C is internalized in *P. gingivalis*-infected macrophages, decreasing the intracellular bacterial load. Intracellular localization of cystatin C in infected macrophages showed that cystatin C co-localizes with *P. gingivalis*. Furthermore, a decrease in the production of inflammatory cytokines and NO was observed. Interestingly, the treatment with cystatin C increases ROS production and decreases the programmed cell death in macrophages infected with the bacterium. These results highlight the important activity of cystatin C against the *P. gingivalis*, in addition to its immunoregulatory role in infected macrophages.

*P. gingivalis* is a keystone pathogen and can impair innate immunity and transform a symbiotic microbiota into a dysbiotic state in periodontal tissue (*Hajishengallis, Darveau & Curtis, 2012*). Conventional treatment against this disease consists of the mechanical elimination of the bacteria by scaling and root planning of the affected tissue (*de Andrade, Almeida-da Silva & Coutinho-Silva, 2019*) and use of bactericides such as chlorhexidine and antibacterials such as metronidazole (*Rosa et al., 2021*; *Larsen, 2002*). However, the clinical application of chlorhexidine is limited by its bitter taste and staining of teeth and

tongue (*Larsen, 2002*). On the other hand, although the use of metronidazole is effective against extracellular *P. gingivalis*, it cannot penetrate infected cells (*Ye et al., 2017*) and various adverse effects have been reported (*Kafadar et al., 2013*).

Antimicrobial peptides arise as promising molecules in the treatment of oral infections such as periodontitis due to their microbicidal effect against *P. gingivalis* and the immunoregulation capacity exerted by them (*Liang & Diana, 2020*). The extracellular antimicrobial activity of cystatin C on various oral microorganisms has been previously reported. Cystatin C inhibits the growth of *Streptococcus mutans* and *Enterococcus faecalis*, through the induction of morphological changes in their cell walls, peptidoglycan, and a decrease in the electron density of the cytoplasm (*Blancas et al., 2021*). *Blankenvoorde et al. (1998)* identified that cystatin C binds to the active site of *P. gingivalis* cysteine protease, inhibiting its proteolytic activity in culture supernatant and decreasing its growth by 50%. Other peptides such as Nal-P-113 and mPE also inhibit *P. gingivalis* at a concentration of 20 μg/mL and 4 μg/mL, respectively (*Wang et al., 2018*; *Hua, Scott & Diamond, 2010*). Our study shows that cystatin C decreases the growth of *P. gingivalis* by 90%, demonstrating that cystatin C has potent antimicrobial activity at low concentrations (2.75 μg/mL) and not cytotoxic effects on macrophages. Cystatin C inhibits growth at a lower concentration than Nal-P-113 and mPE without altering macrophage viability, which suggests cystatin C is an ideal peptide for the treatment of bacteria associated with the oral cavity such as *P. gingivalis*.

Another advantage of antimicrobial peptides such as cystatin C is that, unlike antibiotics such as metronidazole, this peptide is internalized by endocytosis mechanisms in endothelial cells and monocytes (*Singhrao & Olsen, 2018*). Interestingly, in this study, we present several lines of evidence to demonstrate that cystatin C is internalized in *P. gingivalis*-infected macrophages and has intracellular antimicrobial activity against this bacterium. First, by counting colony-forming units, we observed a lowers number of intracellular bacteria in the infected macrophage after treatment with cystatin C. Second, we observed the intracellular localization of cystatin C in infected macrophages and its co-localization with *P. gingivalis*, suggesting that this peptide could have direct activity against the bacteria, favoring its elimination.

*P. gingivalis* internalized in monocytes/macrophages or dendritic cells can thus use a ''*Trojan horse*'' to favor its dissemination to other tissues such as coronary arteries, placenta, brain, and liver, which constitutes a pathogenic mechanism for the development of systemic diseases (*Singhrao & Olsen, 2018*; *Dominy et al., 2019*). In addition, intracellular persistence is a strategy developed by the bacteria to escape the host's immune response, which contributes to the pathological process of periodontitis (*Mei et al., 2020*). In addition to the ability of *P. gingivalis* to invade host cell, bacterium can escape to the autophagosome-lysosome fusion to avoid death (*Werheim et al., 2020*; *Shiheido-Watanabe et al., 2023*). Therefore, the intracellular elimination of this bacterium mediated by treatment with cystatin C constitutes an important finding since it could reduce the presence of *P. gingivalis* in gingival tissue or prevent bacterial dissemination.

In addition to these mechanism, *P. gingivalis* induces the recruitment of inflammatory cell infiltrates, particularly macrophages, which leads to the destruction of supporting tissues (*Darveau, 2009*). Macrophages play an important role in the inflammatory response initiation, due to producing pro-inflammatory cytokines and mediating alveolar bone resorption in periodontitis (*Sun et al., 2021*). Various cytokines produced by macrophages promote the development of periodontitis. The inflammasome activation induces the maturation of two important inflammatory cytokines, IL-1β and IL-18 (*Champaiboon et al., 2014*). In addition, *P. gingivalis* promotes the polarization of M1 macrophages which are directly related to bacterial survival and evasion of the macrophage immune responses in macrophage, thereby inducing the progression of the disease (*Lin et al., 2022*).

We investigated the cystatin C activity in inflammatory cytokines production in *P. gingivalis* infected macrophages. We found that *P. gingivalis* induces the production of IL-1β and TNF-α. However, cystatin C stimulation importantly decreased these cytokines. Interestingly, we found that cystatin C has anti-inflammatory activity due to inducing IL-10 production in macrophages challenged with P. gingivalis. This cytokine production is an important finding in our research because previous reports indicated that IL-10, suppresses TNF-α, IL-6, and IL-1 production (*Sun et al., 2019*). In addition, Il-10 gene polymorphisms may be associated with a predisposition to chronic periodontitis development (*Geng et al., 2018*), which indicates cystatin C could contribute to the inflammatory process decrease by IL-10 production. In a previous study, we demonstrated that cystatin C down-regulates the production of IL-1β and TNF-α in HGFs co-incubated with *P. gingivalis*, whereas IL-10 production increases (*Blancas-Luciano et al., 2022*). This could represent an important mechanism to inhibit an exacerbated inflammatory response in macrophages to *P. gingivalis* infections.

The phenomenon observed in both cell types could suggest that cystatin C is a modulator of inflammatory response in both HGF and macrophages. Cystatin C could be participating in the polarization of M2 macrophages infected with *P. gingivalis*, evidenced by the presence of IL -10. In addition, our findings on the diminished IL-1β and TNF-α production are consistent with the literature, where it is demonstrated that cystatin C down-regulates the production of both cytokines in monocytes stimulated with bacterial LPS (*Gren et al., 2016*). The downregulated production induced by cystatin C can contribute to the limitation of immune damage in periodontal tissues. A balance of inflammatory cytokines and anti-inflammatory cytokines is critical for homeostasis in the host. Therefore, an appropriate immune response must be developed to resist periodontopathic bacteria without excessive damage to periodontal tissues (*Sztukowska et al., 2002*; *Mydel et al., 2006*; *Kikuchi et al., 2005*; *Smalley, Birss & Silver, 2000a*; *Smalley, Birss & Silver, 2000b*). In additionally to cystatin C, other salivary peptides have been described, like histatin 5 and 1, that participate in downregulating inflammatory cytokines, such as TNF-α, IL-1β, IL-6, and IL-8 in macrophages and fibroblasts (*Imatani et al., 2000*; *Lee et al., 2021*).

In addition to cytokine production, macrophages infected with *P. gingivalis* produce microbicidal mediators such as NO and ROS (*Wang et al., 2014*; *Sun et al., 2010*). We found that *P. gingivalis* induces the production of ROS. An additional increase of ROS

was also detected after macrophage stimulation with cystatin C. This finding suggests that the increased ROS production in infected macrophages could represent a microbicidal mechanism exerted by the peptide to reduce the intracellular bacterial load. Antimicrobial peptides, like LL-37, can mediate intracellular bactericidal activity against M. tuberculosis by ROS production in a THP-1 cell line (*Gren et al., 2016*). Additionally, α-defensins LL-37, CG117-136, and protegrins (PG-1) decrease NO and TNF-α production (*Zughaier, Shafer & Stephens, 2005*). Our findings demonstrate that infected macrophages stimulated with cystatin C show an increase of ROS production, together with a decrease of NO, suggesting that ROS production could represent a cystatin C-mediated microbicidal mechanism that helps to reduce the intracellular bacterial load, while NO decrease could inhibit the inflammatory response and tissue damage.

When evaluating NO production in macrophages infected with *P. gingivalis*, we observed an increase in NO production. Interestingly, co-incubation with cystatin C decreased the production of NO importantly. *Ghosh et al. (2008)* demonstrated that NO and its regulatory enzyme iNOS are a key role in the pathogenesis of periodontal disease. Furthermore, diverse studies have reported that high levels of NO and iNOS are relational with periodontal disease severity in patients and rodent models (*Menaka et al., 2009*; *Di Paola et al., 2004*; *Gyurko et al., 2003*).

Nitric oxide can be anti- or pro-apoptotic (*Sharma, Al-Omran & Parvathy, 2007*). It has been reported that NO release by oral keratinocytes infected with *P. gingivalis* may contribute to their resistance to apoptosis. Apoptosis plays a critical role in gingival tissue homeostasis (*Bugueno et al., 2018*). However, chronic inflammatory diseases such as periodontitis, can is associated with tissue destruction. *Listyarifah et al. (2017)* show that in biopsies of patients with periodontitis, there is an expression of caspase-3 in the lamina propria cells and professional phagocytic cells (CD68[+]) (*Listyarifah et al., 2017*).

*P. gingivalis* induces apoptosis on fibroblasts (*Graves et al., 2001*; *Imatani et al., 2004*; *Palm, Khalaf & Bengtsson, 2015*), endothelial cells (*Sheets et al., 2005*), cardiac myoblasts (*Lee et al., 2006*), lymphocytes (*Geatch et al., 1999*), monocytes (*Lee et al., 2006*), and polymorphonuclear neutrophils (*Hiroi et al., 1998*; *Murray & Wilton, 2003*; *Preshaw & Taylor, 2011*). *P. gingivalis* induces apoptosis after invading the host cell, which could contribute to the immune dysregulation observed during chronic periodontitis. *P. gingivalis* induces apoptosis in a time and dose-dependent manner, by regulating pro-apoptotic molecules such as caspase-3, -8, -9, Bid and Bax (*Desta & Graves, 2007*). In this study, we showed that *P. gingivalis* induces apoptosis through the increase of caspase 3 activity in *P. gingivalis*-infected macrophages. In contrast, cystatin C stimulation in infected macrophages reduces cell death. These results agree with the report by *Lee et al. (2006)*, who demonstrated that histatin 5 induces inhibition of apoptosis in human gingival fibroblasts infected with *P. gingivalis*.

## CONCLUSIONS

*P. gingivalis* could induce the recruitment, activation, and disruption of many macrophage functions. These mechanisms contribute to escape macrophage killing and trigger an inflammatory response, which contributes to the periodontitis progression.

Cystatin C is an antimicrobial peptide that can be internalized in macrophages. In addition, this peptide has dual activity, since it inhibits the intracellular bacterial load of *P. gingivalis* and regulates the inflammatory response. The immunoregulatory role was shown through a decrease in inflammatory cytokines and an increase in anti-inflammatory cytokines. Cystatin C decreased NO production and cell apoptosis. The increase in ROS production can explain its potent antimicrobial effect. These findings could aid in a better understanding of this peptide's properties and its possible application in infectious and inflammatory oral diseases.

## ACKNOWLEDGEMENTS

This article is part of the requirements for obtaining a Doctoral degree at the Posgrado en Ciencias Biológicas of Blancas-Luciano Blanca Esther. We thank Adriana Ruiz-Remigio for her technical assistance.

### Funding
This research was funded by PAPITT, DGAPA, UNAM, grants #IN208522. Blanca Esther Blancas-Luciano is supported by CONAHCYT grant #736537 for her doctoral studies. The funders had no role in study design, data collection and analysis, decision to publish, or preparation of the manuscript.

### Grant Disclosures
The following grant information was disclosed by the authors:
PAPITT, DGAPA, UNAM: #IN208522.
CONAHCYT: #736537.

### Competing Interests
The authors declare there are no competing interests.

### Author Contributions

- Blanca Esther Blancas-Luciano conceived and designed the experiments, performed the experiments, analyzed the data, prepared figures and/or tables, authored or reviewed drafts of the article, and approved the final draft.
- Ingeborg Becker-Fauser conceived and designed the experiments, performed the experiments, analyzed the data, authored or reviewed drafts of the article, and approved the final draft.
- Jaime Zamora-Chimal conceived and designed the experiments, performed the experiments, analyzed the data, prepared figures and/or tables, authored or reviewed drafts of the article, and approved the final draft.
- Luis Jiménez-García performed the experiments, authored or reviewed drafts of the article, and approved the final draft.
- Reyna Lara-Martínez performed the experiments, analyzed the data, authored or reviewed drafts of the article, and approved the final draft.

- Armando Pérez-Torres performed the experiments, authored or reviewed drafts of the article, and approved the final draft.
- Margarita González del Pliego analyzed the data, authored or reviewed drafts of the article, and approved the final draft.
- Elsa Liliana Aguirre-Benítez analyzed the data, authored or reviewed drafts of the article, and approved the final draft.
- Ana María Fernández-Presas conceived and designed the experiments, performed the experiments, analyzed the data, prepared figures and/or tables, authored or reviewed drafts of the article, and approved the final draft.

### Human Ethics

The following information was supplied relating to ethical approvals (*i.e.,* approving body and any reference numbers):

Ethics Committee of the Universidad Nacional Autónoma de México.

### Data Availability

Supplemental Information 1

### Supplemental Information

Supplemental information for this article can be found online at http://dx.doi.org/10.7717/peerj.17252#supplemental-information.

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
