# Peer review of "Cystatin C: immunoregulation role in macrophages infected with Porphyromonas gingivalis"

_PeerJ, doi:10.7717/peerj.17252_

## Round 0.1 · original submission · Major Revisions

Please answer point-by-point to the reviewers' comments and questions and address the issues and fix them in the manuscript.

Reviewer 1 ·

Basic reporting

1. Occasionally, there are sections in the text that may come across as somewhat repetitive upon reading. To address this, it could be beneficial for the authors to seek the input of a fluent English speaker for review.
2. I kindly suggest that the authors condense the introduction by focusing on the essential background information necessary for the study.
3. It's not clear what the asterisk at 48 hours in the control group signifies in Figure 1.

Experimental design

1. Could you please provide information regarding the source or origin of the Cystatin C used in all the experiments including immunization? A proof of the specificity of the antibodies obtained should be provided.
2. Could you clarify the differentiation process of monocytes into macrophages? Without growth factors over 48-72 hours, cells typically undergo cell death rather than differentiation into macrophages.
3. In many experiments where cells were infected and exposed to Cystatin C, the MOI (Multiplicity of Infection) and infection duration are not specified. For example, when it is mentioned 'The infected macrophages were stimulated with cystatin C for 24 h,' does this mean the infection occurred simultaneously with Cystatin C treatment?
4. Regarding the antibody production protocol, the injections were administered five times at 8-day intervals, with the first four being intraperitoneal and the last one subcutaneous. Is this correct? Additionally, when referring to 'first and second immunization,' which injections are being referred to?
5. Regarding the confocal microscopy figure: The quality of the image is poor, making it difficult to conclude where Cystatin localizes with respect to bacteria. The meaning of the arrows in the figure is unclear. It would be helpful if the authors could provide a more detailed explanation of what the arrows indicate. In general, It is advisable to include a higher-quality and enlarged version of the figure for better clarity and detail.
6. Could you provide information on the pathway Cystatin C follows to enter the cells?

Validity of the findings

The statement 'cystatin C increases ROS production, which may be related to the antimicrobial activity observed in Figure 1a' would make more sense if the bacteria and cystatin C were located within phagosomes. However, the text consistently mentions that both are localized in the cytoplasm. It would be helpful to clarify what is meant by 'cytoplasm' in this context – whether it refers to the cytosol or organelles. Addressing this aspect is crucial for a better understanding of the study's findings.

·

Basic reporting

Describe inclusion and exclusion criteria for donor patients. Diseases could modify the basal response.

Experimental design

What is the macrophage lysing protocol? Was a buffer used?. What did the buffer contain?.
It would be interesting to add the photos of the plates seeded with the lysed macrophages as a supplementary figure.

Validity of the findings

Quantitatively indicate how much camptothecin and DNase was used as a positive control. Attach reference.

---

## Round 0.2 · Major Revisions

Dear authors, please address the questions from Reviewer 1. Additionally, please consider the following comments to improve the manuscript.

Major comments:

The English language needs improvement throughout the manuscript. There are issues with grammar, sentence structure, word choice, and awkward phrasing. I would suggest having the manuscript edited by a native English speaker.
Examples:

Lines 40-41: "P. gingivalis is a key etiologic agent in periodontitis" should be "P. gingivalis is a key etiological agent of periodontitis."

Line 59: Re-write this sentence for clarity.

Some of the methods lack sufficient detail:
a) In the intracellular survival assay section, provide more details on the calculation of CFU/mL from the cell lysate serial dilutions.

b) For cytokine assays, please state the concentration range of the standards used to generate the calibration curves.

c) For antibody production in mice - include a statement that animal experimentation guidelines were followed and approved by an ethics committee.

There are a few concerns regarding the presentation and interpretation of results:
a) In Figure 1B, statistical significance is indicated, but there appear to be no significant differences in cell viability with cystatin C treatment. Please review the statistics and revise the figures accordingly.

b) In Figure 2, clearly state in the legend which treatment differences have statistical significance.

c) In Figures 4 and 5, specify in the legends which treatment differences are statistically significant, rather than the general statement about significance compared to control groups.

d) The text does not always provide a clear justification for some of the findings, e.g., increased ROS production with cystatin C treatment. Elaborate on the potential mechanisms in the Discussion.

Minor comments:

- Provide information about data availability in a Data Accessibility section.
- Carefully review reference formatting - there seem to be some inconsistencies.
- Avoid excessive use of abbreviations on first usage - spell out the term followed by the abbreviation in parentheses, e.g., Reactive oxygen species (ROS)

To summarize, the manuscript has potential, but it needs some revisions to address the comments mentioned earlier before it can be considered for publication. Additionally, please make sure to have the manuscript reviewed by a native English speaker. I would be glad to review a revised version of it.

I hope you find these comments helpful. Please feel free to contact me if you have any questions.

Sincerely,

CM

Reviewer 1 ·

Basic reporting

see the attached file

Experimental design

see the attached file

Validity of the findings

see the attached file

Annotated reviews are not available for download in order to protect the identity of reviewers who chose to remain anonymous.

·

Basic reporting

No comment

Experimental design

No comment

Validity of the findings

No comment

Additional comments

No comment

---

## Round 0.3 · accepted · Accept

The authors have addressed the previous comments, and the article deserves to be published.